# Convolutional Neural Network with a Topographic Representation Module for EEG-Based Brain—Computer Interfaces

**DOI:** 10.3390/brainsci13020268

**Published:** 2023-02-05

**Authors:** Xinbin Liang, Yaru Liu, Yang Yu, Kaixuan Liu, Yadong Liu, Zongtan Zhou

**Affiliations:** College of Intelligence Science and Technology, National University of Defense Technology, Changsha 410073, China

**Keywords:** convolutional neural network (CNN), electroencephalogram (EEG), topographic representation, brain–computer interface (BCI), EEG decoding, deep learning

## Abstract

Convolutional neural networks (CNNs) have shown great potential in the field of brain–computer interfaces (BCIs) due to their ability to directly process raw electroencephalogram (EEG) signals without artificial feature extraction. Some CNNs have achieved better classification accuracy than that of traditional methods. Raw EEG signals are usually represented as a two-dimensional (2-D) matrix composed of channels and time points, ignoring the spatial topological information of electrodes. Our goal is to make a CNN that takes raw EEG signals as inputs have the ability to learn spatial topological features and improve its classification performance while basically maintaining its original structure. We propose an EEG topographic representation module (TRM). This module consists of (1) a mapping block from raw EEG signals to a 3-D topographic map and (2) a convolution block from the topographic map to an output with the same size as the input. According to the size of the convolutional kernel used in the convolution block, we design two types of TRMs, namely TRM-(5,5) and TRM-(3,3). We embed the two TRM types into three widely used CNNs (ShallowConvNet, DeepConvNet and EEGNet) and test them on two publicly available datasets (the Emergency Braking During Simulated Driving Dataset (EBDSDD) and the High Gamma Dataset (HGD)). Results show that the classification accuracies of all three CNNs are improved on both datasets after using the TRMs. With TRM-(5,5), the average classification accuracies of DeepConvNet, EEGNet and ShallowConvNet are improved by 6.54%, 1.72% and 2.07% on the EBDSDD and by 6.05%, 3.02% and 5.14% on the HGD, respectively; with TRM-(3,3), they are improved by 7.76%, 1.71% and 2.17% on the EBDSDD and by 7.61%, 5.06% and 6.28% on the HGD, respectively. We improve the classification performance of three CNNs on both datasets through the use of TRMs, indicating that they have the capability to mine spatial topological EEG information. More importantly, since the output of a TRM has the same size as the input, CNNs with raw EEG signals as inputs can use this module without changing their original structures.

## 1. Introduction

Brain–computer interfaces (BCIs) enable direct communication between humans and machines via electroencephalogram (EEG) signals [1]. EEG signals contain instinctive biometric information from the human brain. Through precise EEG decoding, BCIs can recognize the inner thoughts of users. The EEG applications in ML/DL-based disease, mental workload and sleep stage prediction have also been widely studied [2,3,4,5,6]. They are also based on the classification of EEG signals and are similar to BCIs in implementation and processing methods. In general, EEG decoding consists of five main stages: data collection, signal preprocessing, feature extraction, classification and data analysis [7]. Although these stages are essentially the same in a BCI paradigm, signal preprocessing [8], feature extraction [9] and classification methods [10] typically require substantial expertise and a priori knowledge about the BCI paradigm. Moreover, due to manual processing, some useful information may be excluded from the extracted features, which poses a challenge in the subsequent classification and data analysis steps.

Deep learning has largely alleviated the need for manual feature extraction with the combination of automatic feature extraction and classification. Convolutional neural networks (CNNs), in particular, have achieved great success in many challenging image classification tasks, outperforming approaches that rely on handcrafted features [11,12]. Inspired by the success of deep learning in areas such as computer vision and natural language processing, researchers have introduced it to the EEG decoding field [13]. CNNs are some of the most versatile deep learning methods in BCI research. Among all deep learning-based EEG decoding methods, those related to CNNs account for 53% of the total (43% for CNNs and 10% for hybrid CNNs) [14]. CNNs are typically composed of three structural blocks: convolutional layers, pooling layers and fully connected layers. A convolutional layer is an essential part of a CNN that performs the feature extraction function. A pooling layer provides a downsampling operation that both ensures the learning of more robust features and reduces the number of required computations. Fully connected layers are typically located at the bottom of the network and implement the combination of local features and classifier functions. The architecture of a CNN generally consists of layers arranged in a specific order, with earlier layers learning lower-level features and deeper layers learning higher-level features. Several studies have used CNN models, including light [15,16,17,18,19,20] and deep [17,21,22,23] architectures, as well as other varieties [24,25,26,27,28,29,30], to decode EEG signals. Waytowich et al. introduced a compact CNN for directly performing feature extraction and classification based on raw steady-state visually evoked potential (SSVEP) signals, with an average cross-subject accuracy of approximately 80% on a 12-class dataset [19]. By introducing batch normalization in the input and convolutional layers to cope with the overfitting problem, Liu et al. applied a CNN to the task of P300 signal detection and achieved state-of-the-art recognition performance on both Dataset IIb of BCI competition II and Dataset II of BCI competition III [20]. Tang et al. proposed a CNN model based on spatial–temporal features to classify single-motor-imagery (MI) tasks, and the results showed that compared with traditional methods, CNNs could further improve the classification performance [23]. To address the overfitting problem faced by traditional machine learning methods in EEG-based emotion reading, Li et al. used a hierarchical CNN with differential entropy features from different channels as inputs and achieved superior classification results compared to those of traditional methods [30]. Li et al. proposed MI-Visual Geometry Group (VGG) by modifying the VGG network to enable the effective recognition of spectral images generated by MI-EEG and obtained competitive results on three publicly available datasets [22].

EEG-based CNNs use both raw signals and features generated from raw signals as inputs. In this paper, we focus only on CNNs with raw 2-D EEG signals as inputs. Raw EEG signals refer to EEG data in the time domain, i.e., the C (channels) × TP (time points) matrix. Since deep learning-based CNN models have the ability to learn complex features from data without using handcrafted features and can achieve end-to-end learning, raw EEG signals are the most commonly used input formulations [31]. Based on their classification errors, CNNs simultaneously learn and optimize the feature representations of raw EEG signals. Several competitive CNN models using raw EEG signals as inputs have been proposed [16,17,23,28,32,33,34,35,36,37]. Schirrmeister et al. more systematically investigated the process of performing end-to-end learning from raw signals in EEG decoding using CNNs and designed two widely used network architectures, DeepConvNet and ShallowConvNet [17]. Test results obtained on two different datasets showed that the proposed networks achieved classification performance that was at least as good as that of the best traditional methods. In addition, a visualization of the learned features also showed that the two networks performed effective spatial mapping. Lawhern et al. introduced a compact CNN, EEGNet, which uses depthwise and separable convolutions. Test results obtained on four different types of BCI paradigms showed that EEGNet had better generalization ability while obtaining comparable classification performance to that of other methods [16]. Amin et al. used raw EEG signals without preprocessing or artifact removal as inputs, and the classification performance was significantly improved by fusing multiple CNN models with different architectures [36]. CNNs with raw EEG signals as inputs ignore the spatial intertopology of the electrodes; therefore, most of these networks contain a spatial (depth) convolutional layer to learn the weights of the electrodes, which is equivalent to a compensatory operation for ignoring the spatial topological EEG information. However, this compensatory operation can only learn some spatial topology at the global level.

In this paper, we introduce a topographic representation module (TRM) to address the spatial topological information loss induced by CNNs with raw EEG signals as inputs. Our goal is to make a CNN that takes raw the EEG signal as input have the ability to learn the topological information in EEG and improve its classification performance while basically maintaining its original structure. The TRM consists of (1) a mapping block from the raw EEG signals to a topographic map and (2) a convolutional block for transforming the topographic map into an output with the same size as the input. That is, the size of the EEG signals remains unchanged after passing through our TRM. CNNs that take raw 2-D EEG signals as inputs can use this module without changing their original network structure. Such a design takes advantage of both the spatial topological information of EEG signals and various excellent existing CNN architectures, making the TRM very versatile. The main contributions of this study are as follows:The TRM is designed, and the dimension and size of EEG signals remain unchanged after passing through the TRM.The TRM can be embedded into a CNN using the raw EEG signals as input without any adjustment to the network structure, which enables TRM to use the existing excellent EEG classification network.The classification results show that with TRM, the results of three commonly used CNNs on two public datasets are better than those of the original network, indicating that TRM has the potential to improve CNN classification performance by learning EEG topological information.

The rest of this paper is organized as follows: Section 2 presents the materials and methods, including the utilized datasets, classification algorithms, TRM, implementation details and evaluation metrics. Section 3 presents the experimental results, i.e., the algorithmic performance achieved in terms of different evaluation metrics. Section 4 presents a discussion of the algorithms and the results. Finally, we provide a conclusion.

## 2. Materials and Methods

### 2.1. Datasets

The EEG signals in BCIs are generally classified into two types according to the presence or absence of external stimuli: evoked potentials and spontaneous EEG signals [7]. Evoked-potential BCIs have clear external stimuli, and their EEG signals exhibit certain time-locked characteristics. Usually, this type of BCI has a high classification accuracy. Unlike evoked-potential BCIs, spontaneous EEG-based BCIs rely on the subject’s spontaneous brain activity, usually without an external stimulus, and are generally more difficult to train. In this paper, we choose an evoked-potential BCI dataset and a spontaneous EEG-based BCI dataset.

The EBDSDD: The EBDSDD has been described in detail in [38]. Here, we briefly describe it as follows: The experiment was carried out on a simulated platform. There were two vehicles in total. One was controlled by a computer as a leading vehicle, with a speed of 100 km/h, and the other was controlled by the subject, following the leading vehicle in front and remaining less than 20 m away from it. The leading vehicle would decelerate randomly in an emergency every 20–40 s, and remind the following vehicle through the brake light. In order to avoid a collision, the subject needed to brake immediately after seeing the brake light of the vehicle in front flashing. The time when the leading vehicle’s brake light came on was marked as a stimulus signal, and the time when the subject started to press the brake pedal was a response signal. The EBDSDD is an evoked-potential EEG dataset containing 2 types of tasks (emergency braking and normal driving) obtained from 18 subjects, each performing approximately 210 emergency braking (target) trials. EEG signals were recorded using 59 electrodes placed on the scalp sites (international 10–20 system, reference at nose), low-pass filtered at 45 Hz and downsampled to 200 Hz. We refer to [38] for the data processing method. Data from 1300 ms before the subjects’ emergency braking actions to 200 ms after braking are chosen as the target segments. The nontarget segments are intercepted from normal driving EEG signals away from any stimulation and braking behavior for at least 3000 ms. Through a sliding window with a length of 1500 ms and a step length of 500 ms, we obtain the normal driving (nontarget) segments. Baseline correction is performed in a segmentwise manner using the data for the first 100 ms. Each subject has about 6628 normal driving (nontarget) trials. For each subject, we choose the same number of nontarget segments and target segments. Therefore, the number of emergency braking (target) and normal driving (nontarget) trials for each subject is about 210. In addition, electrodes FP1, FP2, AF3 and AF4 are susceptible to interference from the oculomotor potential, so we exclude these and use the remaining 55 electrodes. Therefore, the size of each target and nontarget segment is 55 (channels) × 280 (time points).

The HGD: The HGD has been described in detail in [17]. Here, we give a brief description. The HGD is a spontaneous EEG dataset containing 4 classes of movements (left hand, right hand, feet and rest) obtained from 14 healthy subjects, each possessing approximately 1040 trials of executed movements with lengths of 4 s. The experiment has 13 runs in total, and each run contains 80 movement cues. The order of presentation is pseudo-random, and every 4 trials show all the 4 types of movement cues. Ideally, each type of movement contains 260 trials. The first approximately 880 trials are the training set, and the last approximately 160 trials are the test set. The HGD is a 128-electrode dataset with a sampling rate of 500 Hz. For the data processing method, we refer to [17]. Forty-four electrodes covering the motor cortex (all central electrodes except Cz, which is used as the recording reference electrode) are selected. The EEG signals are filtered using a 4–125 Hz bandpass filter and downsampled to 250 Hz. We adopt the standard trialwise training strategy, using the entire duration of each trial, so the data matrix size for each trial is 44 (channels) × 1000 (time points).

### 2.2. Classification Algorithms

In this paper, we use three representative and widely used EEG-based CNNs: ShallowConvNet, DeepConvNet and EEGNet. They are CNNs specifically designed for EEG decoding tasks, and they have achieved better performance than traditional methods in many BCI applications.

ShallowConvNet: The design of ShallowConvNet is inspired by the filter bank common spatial pattern (FBCSP). It is similar to the FBCSP in terms of its EEG feature extraction process. ShallowConvNet has a simple architecture, consisting only of a temporal convolutional layer, a spatial convolutional layer, an average pooling layer and a dense classification layer. It has achieved better results than the best traditional methods in many EEG decoding tasks [16,17,21,39,40,41,42]. For more details about ShallowConvNet, please refer to [17].

DeepConvNet: The structure of DeepConvNet is inspired by the successful architectures of deep CNNs in computer vision; it aims to extract a wide range of features without relying on specific feature types. DeepConvNet is designed as a general CNN architecture with the hope of achieving competitive accuracy by using only a small amount of expert knowledge. It consists of four convolutional pooling blocks and a dense classification layer. The first convolutional pooling block is divided into a temporal convolutional layer, a spatial convolutional layer and a max pooling layer. The remaining convolutional pooling blocks consist of only one convolutional layer and one max pooling layer. DeepConvNet has achieved competitive classification accuracy compared to that of traditional methods in many EEG decoding tasks [16,17,21,39,41,42,43]. For more details about DeepConvNet, please refer to [17].

EEGNet: EEGNet is designed to find a single CNN architecture that can be applied to different types of EEG-based BCIs and make the network as compact as possible. The structure of EEGNet consists of 2 convolutional pooling blocks and a classification layer. Block 1 performs a temporal convolution and a depthwise convolution sequentially. Block 2 uses a separable convolution consisting of a depthwise convolution and a pointwise convolution. In the classification layer, the softmax method is used. Due to the use of depthwise and pointwise convolutions and the omission of the dense layer, this design reduces the number of trainable parameters by at least one order of magnitude relative to other CNNs. Related studies have shown that EEGNet has a reasonable structure and excellent performance in different types of BCI paradigms [16,21,39,40,41,42,43,44,45,46,47,48]. For a detailed description of EEGNet, please refer to [16].

### 2.3. TRM

To make the CNNs with raw EEG signals as inputs more effective in utilizing the spatial topological electrode information and to take advantage of various excellent EEG-based CNNs, we propose the EEG TRM. This module consists of (1) a mapping block from the raw EEG signals to a 3-D topographic map and (2) a convolution block from the 3-D topographic map to the processed output with the same size as the input, as shown in Figure 1.

Mapping block: According to the correspondence between the channels and electrode locations on the scalp, the raw EEG signals with a total size of C × TP are mapped into a 3-D EEG topographic map with a size of H (height) × W (width) × TP (time points). For the correspondence between the electrodes and the 2-D matrix coordinates with a size of H × W, we refer to [49,50] and adjust the size of the matrix according to the electrode distribution. Figure 2 shows the correspondence between the electrode locations and 2-D matrix coordinates for the EBDSDD and the HGD, respectively. The values of the yellow points in each matrix are the potential values of the corresponding electrodes, and the values of the gray points are set to 0. For the reason why the matrix coordinate without electrode mapping is assigned to 0, we refer to the practice of [50], and in the process of designing the TRM, we also found that setting to 0 can achieve better classification results than using interpolation operation. According to the number and distribution of electrodes used, we can adjust the size of the mapped matrix accordingly. The EEG signal at any time can be mapped into a 2-D matrix according to the corresponding relationship between the electrode and the matrix; that is, each matrix is a 2-D EEG topographic map for that moment. After mapping, the potential values of the raw EEG signals remain unchanged in the 2-D matrix, while for those coordinates without corresponding electrodes in the matrix, their values are set to 0. In this way, we transform the raw EEG signals at a certain time into a 2-D topographic map represented by electrode locations (matrix coordinates) and their corresponding potentials. These 2-D topographic maps are combined in temporal order to form a 3-D topographic map.

Convolution block: We perform convolution on the 3-D EEG topographic map to take advantage of its powerful feature learning capability. In addition, to take advantage of the various excellent EEG-based CNN architectures, we make the output of the TRM the same size as the input so that it can be embedded into these networks. Previous studies have shown that the size of the convolution kernel can affect the classification performance of neural networks. According to the sizes of the convolutional kernels used in the convolutional blocks, we design two types of TRMs, namely TRM-(5,5) and TRM-(3,3). In TRM-(5,5), the first layer uses C convolution kernels of size 5 × 5 with a step size of 1. The sizes of the convolution kernels in the subsequent layers are chosen according to the size of the feature map obtained after convolution in the previous layer. If the feature map is larger than 5 × 5, C convolution kernels of size 5 × 5 are used; otherwise, C convolution kernels with the same size as the feature map are used. Similarly, in TRM-(3,3), the first layer uses C convolution kernels of size 3 × 3 with a step size of 1. The sizes of the convolution kernels in the subsequent layers are chosen according to the size of the feature map obtained after convolution in the previous layer. If the feature map is larger than 3 × 3, C convolution kernels of size 3 × 3 are used; otherwise, C convolution kernels with the same size as the feature map are used. Thus, the 3-D EEG topographic map is convolved to produce an output of size C × TP, which is the same size as that of the input. After all convolutional layers, batch normalization is used to adjust the distribution of the data and speed up the training process.

The input and output of the TRM have the same size, that is, the size of the EEG signal remains unchanged after passing through the TRM, so it can be directly embedded in the front end of the CNN without any structural adjustment to the original network. Meanwhile, the internal structure of TRM is composed of mapping, convolution, batch normalization and resizing operations, which are common operations of CNNs, making it possible to conduct joint training with the CNN. Table A1 and Table A2 show the structure of TRM on the EBDSDD and HGD, respectively. The size of the 2-D matrix in the mapping block can be adjusted according to the number and distribution of electrodes used. For example, on the EBDSDD, we use a 7 × 9 matrix, while on the HGD, we use a 7 × 7 matrix. For the commonly used 64-lead electrodes, the size of the mapping matrix is usually not greater than 9 × 9. Therefore, we use the commonly used 3 × 3 and 5 × 5 convolution kernels. We choose C (number of channels) convolution kernels to ensure that the EEG signal can remain the same size after TRM.

### 2.4. Implementation Details

Figure 2 shows the correspondence between the electrode locations and 2-D matrix coordinates. The upper and lower panels show the correspondence between the electrode locations and matrix coordinates for the EBDSDD and HGD, respectively. Through the mapping method in TRM, the raw EEG signal can be mapped into the corresponding 2-D matrix. Each matrix is a 2-D EEG topographic map for that moment. By arranging these 2-D matrices according to the temporal orders of the EEG signals, a 3-D EEG topographic map is formed, as shown in Figure 1. Both datasets are used to train and test DeepConvNet, ShallowConvNet and EEGNet in their original forms and with TRM-(5,5) or TRM-(3,3) for each subject. Table A1 and Table A2 show the TRM architecture on the EBDSDD and HGD, respectively.

For the division of training set, verification set and test set, we adopt different methods according to the difference between the two datasets. For the EBDSDD, we use a 4-fold cross-validation approach, with 50% of each subject’s data as the training set, 25% as the validation set and the remaining 25% as the test set. For the HGD, the original data for each subject have been divided into a training set and a test set. We keep the test set unchanged and randomly select 80% of the original training set as the new training set and the remaining 20% as the validation set, and each subject’s data are used for 4 training and testing rounds, and the average is taken as the final result. For a fair comparison, we set the same training and testing conditions for all three CNNs and their varieties after using the TRM. For each algorithm, the settings are the same except for the use or nonuse of the TRM. All algorithms are trained and tested on a computer with an NVIDIA 2080 Ti graphics card.

Several aspects of the algorithms are set up as follows:The Adam optimizer is used with a weight decay of 0.001, and the remaining parameters are set to their default values.The cross-entropy loss is used as a criterion.The batch size is set to 32.The number of training epochs is set to 300, and the minimum validation loss strategy is used. The network is trained on the training set and validated on the validation set. After each training, the loss on the validation set is compared with the previous loss, and the model with reduced validation loss is saved. The algorithm model with the lowest loss on the validation set is saved for testing.Both ShallowConvNet and EEGNet are trained and tested on both datasets using the codes officially provided by BrainDecode [17]. On the HGD, we use the code of DeepConvNet provided by BrainDecode, while on the EBDSDD, we adjust DeepConvNet with the settings recommended by [39] since the size of the input data does not meet the minimum length requirement of the original DeepConvNet.

### 2.5. Evaluation Metrics

We comprehensively compare these algorithms using metrics such as classification accuracy, training loss, validation loss, the number of training epochs leading to the lowest validation loss, and a time consumption analysis. Classification accuracy refers to the ratio of the number of correct classifications to the total number of classifications. The training loss and validation loss are the cross-entropy losses of the algorithms on the training and validation sets, respectively. We also count the number of training epochs that leads to the lowest validation loss for each algorithm with the aim of determining which algorithm converges most easily. For the time consumption analysis, we use the time required for training and validation for 300 epochs with each algorithm.

## 3. Results

### 3.1. Classification Accuracy

Table 1 shows the classification accuracies obtained by three CNNs (ShallowConvNet, DeepConvNet and EEGNet) on the EBDSDD with and without the TRM. The results for each subject are the mean value ± standard deviation of 4-fold cross-validation, and the *p* values in the table are calculated by a two-tailed paired *t* test. The average classification accuracies of DeepConvNet, DeepConvNet-TRM-(5,5) and DeepConvNet-TRM-(3,3) are 84.21%, 90.75% and 91.97%, respectively; for EEGNet, EEGNet-TRM-(5,5) and EEGNet-TRM-(3,3), the results are 93.39%, 95.11% and 95.10%, respectively; for ShallowConvNet, ShallowConvNet-TRM-(5,5) and ShallowConvNet-TRM-(3,3), the results are 93.21%, 95.28% and 95.38%, respectively. For DeepConvNet, the classification accuracies are improved for all 18 subjects after using TRM-(5,5), with a maximum improvement of 22.60% (Subject VPbba), and the average classification accuracy is improved by 6.54% (*p* < 0.001). The classification accuracies of 17 subjects are improved after using TRM-(3,3), with a maximum increase of 27.40% (Subject VPbba); only one subject has a slight decrease (Subject VPsaj with −0.46%), and the average classification accuracy is improved by 7.76% (*p* < 0.001). For EEGNet, after using TRM-(5,5), the classification accuracies of 17 subjects are improved, with a maximum increase of 2.92% (Subject VPbad); only one subject shows a slight decrease (Subject VPgaa with −0.42%), and the average accuracy is improved by 1.72% (*p* < 0.001). With TRM-(3,3), the classification accuracies of 15 subjects are improved, with a maximum increase of 4.17% (Subject VPbax); three subjects exhibit a decline (Subject VPgaa with −0.85%, Subject VPgab with −0.23% and Subject VPgal with −0.99%), and the average accuracy is improved by 1.71% (*p* < 0.001). For ShallowConvNet, after using TRM-(5,5), the classification accuracies increase for all 18 subjects with a maximum increase of 5.82% (Subject VPbba), and the average result is improved by 2.07% (*p* < 0.001); after using TRM-(3,3); the classification accuracies increase for 15 subjects with a maximum increase of 6.17% (Subject VPbba), decrease for 2 subjects (Subject VPbax with −0.44% and Subject VPja with −0.72%), and remain unchanged for 1 subject (Subject VPgab), and the average result is improved by 2.17% (*p* < 0.001).

Table 2 shows the classification accuracies achieved by the three CNNs on the HGD with and without the TRM. The average classification accuracies of DeepConvNet, DeepConvNet-TRM-(5,5) and DeepConvNet-TRM-(3,3) are 67.09%, 73.14% and 74.70%, respectively; for EEGNet, EEGNet-TRM-(5,5) and EEGNet-TRM-(3,3), the results are 76.48%, 79.50% and 81.54%, respectively; for ShallowConvNet, ShallowConvNet-TRM-(5,5) and ShallowConvNet-TRM-(3,3), the results are 81.83%, 86.97% and 88.11%, respectively. For DeepConvNet, the classification accuracies increase for 11 subjects after using TRM-(5,5), with a maximum increase of 22.33% (Subject 9), and decrease for 3 subjects (Subject 2 with −0.94%, Subject 11 with −2.66% and Subject 12 with −2.03%), and the average accuracy is improved by 6.05% (*p* < 0.01). After using TRM-(3,3), the classification accuracies increase for 12 subjects, with a maximum increase of 22.49% (Subject 9), and decrease for 2 subjects (Subject 10 with −2.82% and Subject 12 with −1.25%); the average result increases by 7.61%. For EEGNet, 13 subjects exhibit accuracy increases after using TRM-(5,5), with a maximum increase of 10.63% (Subject 6); one subject shows a decrease (Subject 9 with −7.81%), and the average accuracy is improved by 3.02% (*p* < 0.05). All 14 subjects exhibit classification accuracy increases after using TRM-(3,3), with a maximum increase of 11.72% (Subject 5), and the average classification accuracy is improved by 5.06% (*p* < 0.001). For ShallowConvNet, after using TRM-(5,5), the classification accuracies increase for 13 subjects with a maximum of 18.44% (Subject 11) and decrease for 1 subject (Subject 9 with −1.87%); the average classification accuracy is improved by 5.14% (*p* < 0.01). After using TRM-(3,3), the classification accuracies increase for 13 subjects with a maximum of 17.03% (Subject 11) and decrease for 1 subject (Subject 3 with −0.16%), and the average classification accuracy is improved by 6.28% (*p* < 0.001).

### 3.2. Training Loss

Figure 3 shows the average training cross-entropy loss curves for each algorithm. Algorithms with “-TRM” use the TRM. The left and right panels show the average training loss curves for 18 subjects on the EBDSDD and 14 subjects on the HGD, respectively. With the TRM, the training loss curves of all three CNNs follow roughly the same trends as their original forms on both datasets. DeepConvNet and its varieties have the fastest declines in their training loss curves, followed by ShallowConvNet, while the curves of EEGNet and its varieties are relatively flat. On the EBDSDD, the losses of all algorithms are relatively small after a period of training, which is related to the fact that this dataset is a two-classification task with a high classification accuracy. We find that the training loss curves are not smooth, especially those of DeepConvNet, ShallowConvNet and their varieties on the HGD, which is related to the use of weight decay. By imposing certain restrictions on the learning weights, the overfitting problem can be mitigated to some extent. The training loss curves of EEGNet and its varieties show relatively smooth downward trends on both datasets, which we believe is related to the fact that it has fewer trainable parameters than the other two algorithms.

### 3.3. Validation Loss

Figure 4 shows the average validation loss curves obtained by each algorithm. Compared with the training loss curve, the validation curve provides a more accurate reflection of the classification and convergence performance of an algorithm. Similar to what we see in the training loss curves, with either TRM-(5,5) or TRM-(3,3), the validation loss curves of all three CNNs have similar trends to those of their original forms on both datasets. After the first few epochs of training, the validation loss of DeepConvNet is significantly higher than those of the other two algorithms on both datasets, and even with the TRM, its validation loss is still higher than those of the other two methods; this result is consistent with the classification accuracy. On the EBDSDD, EEGNet, EEGNet-TRMs, ShallowConvNet and ShallowConvNet-TRMs all have good validation loss curves, while on the HGD, ShallowConvNet and ShallowConvNet-TRMs are better than EEGNet and EEGNet-TRMs. With the TRM, the validation loss curves of all three algorithms show decreasing trends, indicating that the TRM has the ability to improve classification performance.

### 3.4. The Number of Training Epochs That Yields the Lowest Validation Loss

Figure 5 shows the average number of training epochs that yields the lowest validation loss for each algorithm. The average number of epochs required on the EBDSDD is overall higher than that needed for the HGD. With TRM-(5,5), the average number of training epochs yielding the lowest validation loss decreases for DeepConvNet (*p* = 0.25) and ShallowConvNet (*p* = 0.22) and increases for EEGNet (*p* = 0.09) on the EBDSDD; on the HGD, the average number decreases for DeepConvNet (*p* = 0.07) and EEGNet (*p* = 0.72) and increases for ShallowConvNet (*p* = 0.06). With TRM-(3,3), the average number of training epochs required for the lowest validation loss increases for DeepConvNet (*p* = 0.87), EEGNet (*p* = 0.13) and ShallowConvNet (*p* = 0.46) on the EBDSDD; on the HGD, the average number decreases for DeepConvNet (*p* = 0.29) and increases for EEGNet (*p* = 0.23) and ShallowConvNet (*p* = 0.02). The *p* values are calculated by two-tailed paired *t* tests.

### 3.5. Time Consumption Analysis

Considering the practicality of BCIs, we need to consider not only the classification accuracy but also the execution time of each algorithm. Due to the addition of the TRM, the execution time of each algorithm definitely increases. Figure 6 shows the average time consumption levels of the algorithms for 300 training and validation epochs on each subject. On both datasets, ShallowConvNet consumes less time than DeepConvNet, and DeepConvNet consumes less time than EEGNet. With TRM-(5,5), the times required by ShallowConvNet, DeepConvNet and EEGNet for conducting training and validation for 300 epochs increase by approximately 86%, 48% and 43% on the EBDSDD and approximately 49%, 32% and 21% on the HGD, respectively. With TRM-(3,3), the times required by ShallowConvNet, DeepConvNet and EEGNet increase by approximately 158%, 89% and 67% on the EBDSDD and approximately 111%, 74% and 41% on the HGD, respectively. The execution time of ShallowConvNet-TRM-(5,5) is comparable to that of DeepConvNet, while that of DeepConvNet-TRM (5, 5) is less than that of EEGNet. ShallowConvNet-TRM-(3,3) has shorter execution times than EEGNet on both datasets, while DeepConvNet-TRM-(3,3) has a greater execution time than EEGNet on the EBDSDD and a smaller time on the HGD. The increased time consumption is relatively acceptable.

## 4. Discussion

A raw EEG signal is generally represented as a 2-D matrix form with C (channels) × TP (time points). In a CNN with raw EEG signals as inputs, if there is no representation module concerning the spatial topology of electrodes in the network, the EEG signals are processed as tensors similar to 2-D pictures, and the spatial topological information of the electrodes is ignored. A topographic map is a representation of an EEG signal as a 2-D or 3-D image, depending on the spatial topology of the electrodes (their locations on the scalp) [31]. Topographic maps can be constructed using either raw EEG signals [24,51,52] or extracted features [18,49,50,53,54]. If a CNN uses the EEG topographic map constructed from the extracted features as input, its performance depends on the quality of the utilized features, which often requires substantial expertise and a priori knowledge. In studies involving CNNs using topographic maps constructed from raw EEG signals, to the best of our current knowledge, all the networks were designed according to the needs of the associated tasks. In References [49,50,53], the generation process from the international 10–20 system to a 9 × 9 2-D matrix is described. The generation process not only basically preserves the relative position of the electrode on the scalp, but also maps the irregular electrode distribution into a regular 2-D matrix form, which can be seen as a form similar to the image, making it very suitable for processing using CNNs. Given that there are already many excellent CNNs that use raw EEG signals as inputs, it is possible to achieve the purpose of using the spatial topological information of EEG signals by simply adding a module without changing the structure of the original network.

Therefore, we design the TRM. By mapping raw EEG signals into a 3-D topographic map, we make the input contain the spatial topological information of the electrodes. For the points without corresponding electrodes, we adopt the practice used in [50] and directly set them to 0 instead of using interpolation because, in our experiments, we find that interpolation does not lead to performance improvement but instead increases the time consumption. We perform a convolution operation on the 3-D EEG topographic map to transform it into an output with the same size and dimensions as the input. Depending on the size of the convolutional kernels used, two convolutional strategies, TRM-(5,5) and TRM-(3,3), are chosen to analyze the impact of the size of the convolutional kernels on the classification performance of the employed algorithm. Such a design takes advantage of the powerful feature learning capabilities of deep learning while using a variety of existing excellent CNNs in EEG-based BCIs.

We choose three widely used CNNs, namely ShallowConvNet with a shallow structure, DeepConvNet with a deep structure, and the compact EEGNet. We use two datasets derived from different types of BCI paradigms. It is hoped that these practices make our results representative. We are very pleased to find that each CNN utilizing the TRM exhibits a strong similarity to its original network, in terms of both the training loss and the validation loss curves, indicating that the properties of the original CNN are largely preserved with the use of the TRM. Under the same training and test conditions, the validation loss curves of all three CNNs exhibit downward trends after using TRM-(5,5) or TRM-(3,3) on both datasets, which to some extent indicates the ability of the TRM to improve the classification performance of EEG-based CNNs.

As the size of the 3-D topographic map obtained from the raw EEG signals and the size of the utilized convolutional kernel vary, the number of trainable parameters in the TRM changes accordingly. Figure 7 shows the numbers of trainable parameters in the three CNNs with or without the TRM. In this paper, the number of trainable parameters for TRM-(5,5) is 46,860 when used on the EBDSDD and 18,612 on the HGD, and the number of trainable parameters for TRM-(3,3) is 64,130 when used on the EBDSDD and 35,332 when used on the HGD.

ShallowConvNet has a simple structure, consisting of only a temporal convolutional layer, a spatial convolutional layer, a pooling layer and a dense layer, and its number of parameters is relatively moderate, so it performs the fastest among all three CNNs. Even with TRM-(5,5), the time it consumes is still comparable to that of DeepConvNet and significantly less than that of EEGNet. When using TRM-(3,3), it consumes more time than DeepConvNet but still less time than EEGNet. On the EBDSDD, ShallowConvNet and its varieties achieve the best results on 10 out of 18 subjects, with ShallowConvNet-TRM-(5,5) accounting for 6 and ShallowConvNet-TRM-(3,3) accounting for 4. The average classification accuracy of ShallowConvNet is lower than that of EEGNet, while the accuracies of ShallowConvNet-TRM-(5,5) and ShallowConvNet-TRM-(3,3) are higher than that of EEGNet. Considering the classification accuracy and the execution time of the algorithm, we recommend ShallowConvNet-TRM-(5,5). On the HGD, ShallowConvNet and its varieties achieve the highest classification accuracy on 12 out of 14 subjects, with ShallowConvNet-TRM-(5,5) and ShallowConvNet-TRM-(3,3) each accounting for 6 of them, and ShallowConvNet-TRM-(3,3) achieves the highest average classification accuracy. Inspired by FBCSP, ShallowConvNet often has a better classification performance in spontaneous EEG decoding [16,39]. From the validation loss curves yielded on the HGD, we can also find that ShallowConvNet, ShallowConvNet-TRM-(5,5) and ShallowConvNet-TRM-(3,3) have smaller validation losses than the other methods. Thus, ShallowConvNet-TRM-(3,3) is the recommended algorithm to be used on the HGD.

DeepConvNet has a relatively deep structure and a large number of trainable parameters, so it often requires a large number of samples and uses certain skills for training. With fewer samples, it is often prone to overfitting [16,21,39]. The classification results of DeepConvNet are poor on both datasets, and the validation loss curves also indicate that it has a larger validation loss than the other methods. Although the classification accuracy of DeepConvNet is improved after using the TRM, it is still lower than that of ShallowConvNet and EEGNet, which may be related to the small number of samples. The time consumption of DeepConvNet is higher than that of ShallowConvNet. With TRM-(5,5), its time consumption increases but is still lower than that of EEGNet. When TRM-(3,3) is used, its time consumption exceeds that of EEGNet on the EBDSDD and remains lower than that of EEGNet on the HGD. DeepConvNet and its varieties are not recommended when the trainable sample size is small.

By using depthwise and separable convolutions and omitting the dense layer, the number of trainable parameters in EEGNet is at least one order of magnitude smaller than that in the other two algorithms, which greatly alleviates the overfitting problem that often occurs in deep learning [16]. Compared to the other two methods, EEGNet has smoother training loss and validation loss curves, and the same is true when the TRM is used. Although EEGNet has the fewest trainable parameters, it has the highest time consumption among the three CNNs. With the TRM, its trainable parameter quantity is significantly improved. On the EBDSDD, EEGNet and its variants with the TRM achieve the best results in 8 out of 18 subjects, with EEGNet, EEGNet-TRM-(5,5) and EEGNet-TRM-(3,3) accounting for 1, 2 and 5, respectively. Compared with that of EEGNet, the average classification accuracy of EEGNet-TRM-(5,5) is improved by 1.72% (*p* value < 0.001), which is very valuable. On the HGD, EEGNet and its varieties achieve the best results for 2 out of 14 subjects, and both of them are achieved by EEGNet-TRM-(3,3). The average classification accuracy of EEGNet with TRM-(3,3) is improved by 5.06% (*p* < 0.001), but it is still lower than that of ShallowConvNet.

We also notice that TRM has some disadvantages. As another EEG topographic representation module is added at the input end of the CNN, the newly generated network will inevitably increase the number of trainable parameters and execution time, as shown in Figure 6 and Figure 7. At the same time, we also found that when TRM was used for cross-subject training and testing, its performance was not significantly improved. We believe that this may be related to the obvious individual differences in EEG signals, and the topological information obtained from training for a single individual is not applicable to other individuals. The successful transfer of the model trained in one subject to other subjects requires further research. By analyzing the TRM structure, we find that a large portion of the time consumption is spent on mapping the raw EEG signals into the 3-D topographic map. If the EEG signals are collected as a 3-D matrix (3-D topographic map), the TRM can omit the mapping process. In this case, the additional time consumption caused by the TRM will be very little.

## 5. Conclusions

In this paper, we introduce the TRM. This module consists of a mapping block and a convolution block, and its output has the same size as the input. CNNs that take raw EEG signals as inputs can use the TRM without changing their structure. We select three representative CNNs and test them on two different types of datasets. The results show that the classification accuracies of all three CNNs are improved with the TRM. Next, we intend to use the TRM on more CNNs and datasets for further validation.

## Figures and Tables

**Figure 1 brainsci-13-00268-f001:**
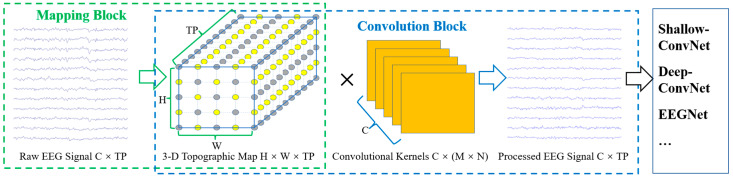
Overall visualization of the TRM. C: channels, TP: time points, H: height, W: width, (M × N): kernel size. The parts framed by green and blue dashed lines are the mapping block and the convolution block, respectively. In the 3-D topographic map, the coordinates of the yellow points correspond to the electrode locations of the raw EEG signals, and the values are the potential values of the electrodes, while the gray points have no corresponding electrodes, and their values are set to 0. The outputs and inputs of the TRM possess the same size.

**Figure 2 brainsci-13-00268-f002:**
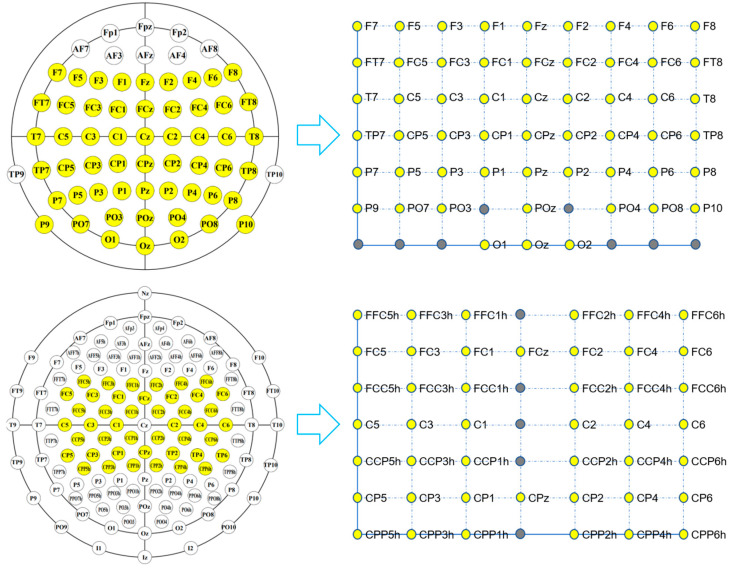
Correspondence between the electrode locations and matrix coordinates. The upper and lower panels show the correspondence between the electrode locations and matrix coordinates for the EBDSDD and HGD, respectively. The values of the yellow points in the matrix are the potential values of the corresponding electrodes, and values of the gray points are set to 0. For the EBDSDD, a total of 55 electrodes are used, and the size of the corresponding matrix is 7 × 9. For the HGD, a total of 44 electrodes covering the motor cortex (all central electrodes except Cz, which is used as the recording reference electrode) are used, and the size of the corresponding matrix is 7 × 7.

**Figure 3 brainsci-13-00268-f003:**
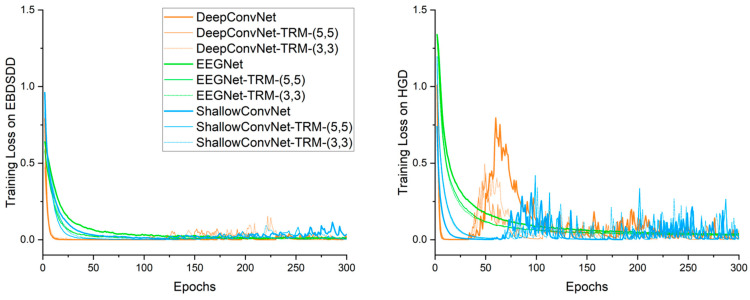
Training cross-entropy loss curves of different algorithms. The left and right panels show the average training loss curves for 18 subjects on the EBDSDD and 14 subjects on the HGD, respectively. Algorithms with “-TRM” use the TRM.

**Figure 4 brainsci-13-00268-f004:**
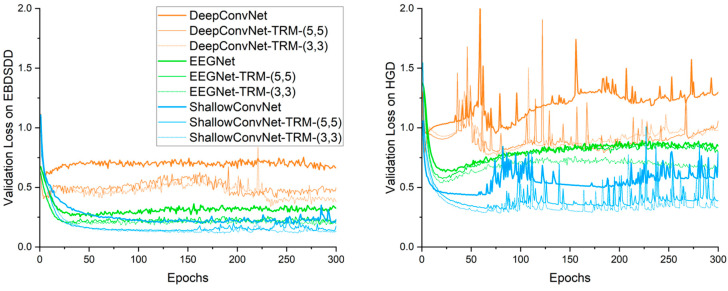
Validation cross-entropy loss curves of different algorithms. The left and right panels show the average validation loss curves on the EBDSDD and HGD, respectively. Algorithms with “-TRM” use the TRM.

**Figure 5 brainsci-13-00268-f005:**
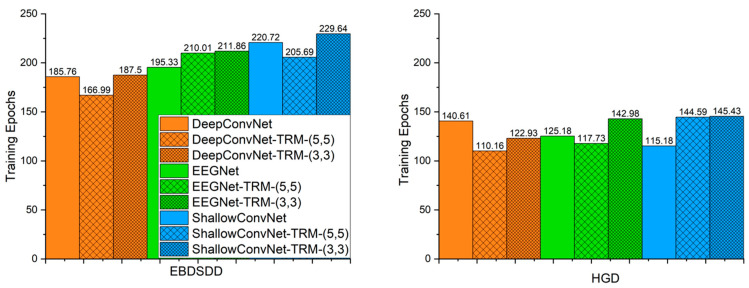
The average number of training epochs yielding the lowest validation loss for each algorithm. The left and right panels show the average numbers of training epochs yielding the lowest validation loss on the EBDSDD and HGD, respectively. Algorithms with “-TRM” use the TRM.

**Figure 6 brainsci-13-00268-f006:**
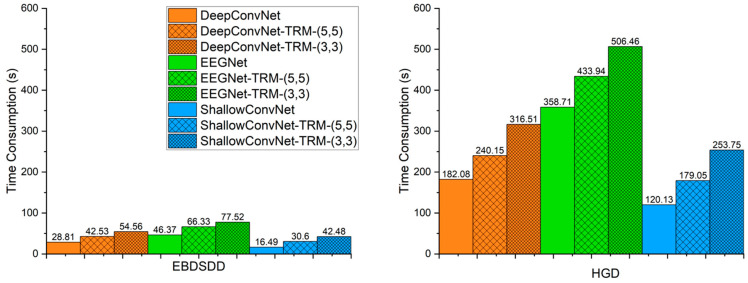
Average time consumption required for training and validation for 300 epochs of algorithms. The left and right panels show the average time consumption on the EBDSDD and HGD, respectively. Algorithms with “-TRM” use the TRM.

**Figure 7 brainsci-13-00268-f007:**
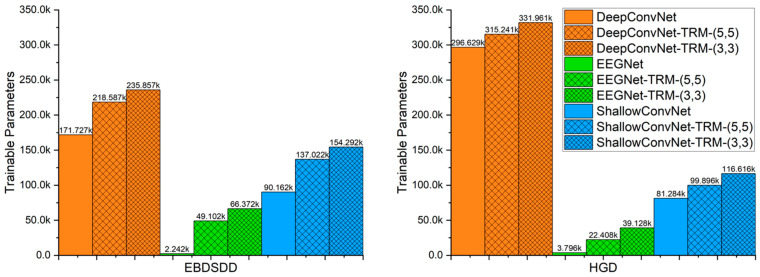
The numbers of trainable parameters for each algorithm. The left and right panels show the numbers of trainable parameters of the algorithms when used on the EBDSDD and HGD, respectively. Algorithms with “-TRM” use the TRM.

**Table 1 brainsci-13-00268-t001:** Classification accuracies achieved by different algorithms on the EBDSDD.

Subject	DeepConvNet			EEGNet			ShallowConvNet		
	Original	TRM-(5,5)	TRM-(3,3)	Original	TRM-(5,5)	TRM-(3,3)	Original	TRM-(5,5)	TRM-(3,3)
VPae	75.82 ± 4.10	84.78 ± 4.78	82.61 ± 3.20	90.22 ± 2.35	92.93 ± 4.21	94.02 ± 3.26	86.96 ± 3.97	91.85 ± 2.59	91.85 ± 2.08
VPbad	88.51 ± 2.79	95.27 ± 2.88	96.17 ± 1.35	95.05 ± 2.14	97.97 ± 1.35	97.97 ± 1.54	93.02 ± 4.79	97.75 ± 2.27	98.65 ± 0.52
VPbax	83.77 ± 2.09	91.89 ± 2.99	93.64 ± 1.95	90.57 ± 2.52	92.32 ± 1.50	94.74 ± 1.89	94.74 ± 1.60	95.39 ± 1.81	94.30 ± 1.68
VPbba	55.82 ± 9.58	78.42 ± 3.24	83.22 ± 7.53	88.70 ± 4.53	91.44 ± 1.72	91.44 ± 3.60	84.93 ± 4.88	90.75 ± 1.72	91.10 ± 2.37
VPdx	81.19 ± 3.70	91.09 ± 3.13	96.78 ± 1.69	91.83 ± 3.74	94.55 ± 0.99	93.32 ± 2.20	93.32 ± 1.49	96.29 ± 3.47	97.03 ± 2.91
VPgaa	94.70 ± 3.81	97.67 ± 1.07	98.09 ± 1.07	98.94 ± 0.42	98.52 ± 0.42	98.09 ± 1.60	97.25 ± 1.45	98.52 ± 0.81	98.09 ± 0.81
VPgab	88.89 ± 2.51	93.98 ± 1.60	94.21 ± 2.19	94.91 ± 1.93	96.30 ± 0.76	94.68 ± 2.87	96.99 ± 1.58	97.92 ± 1.17	96.99 ± 1.17
VPgac	91.52 ± 3.14	95.09 ± 0.89	95.54 ± 2.19	96.88 ± 1.15	97.77 ± 0.89	97.77 ± 0.52	97.32 ± 1.63	97.99 ± 1.69	97.99 ± 1.12
VPgae	87.28 ± 3.69	88.82 ± 4.14	89.47 ± 3.43	90.57 ± 4.38	92.98 ± 2.48	91.89 ± 4.01	86.84 ± 2.77	87.28 ± 2.09	91.45 ± 3.68
VPgag	89.95 ± 4.19	96.81 ± 2.70	98.04 ± 0.80	95.83 ± 2.58	98.04 ± 1.39	98.04 ± 1.39	95.83 ± 1.47	97.06 ± 0.80	97.79 ± 0.94
VPgah	81.45 ± 4.92	86.56 ± 4.16	89.52 ± 7.97	91.13 ± 2.22	92.47 ± 3.04	92.74 ± 3.09	91.13 ± 2.69	93.55 ± 2.15	91.94 ± 1.39
VPgal	80.69 ± 2.97	84.90 ± 1.87	89.60 ± 2.49	93.81 ± 2.73	94.06 ± 0.81	92.82 ± 2.04	94.31 ± 1.25	94.80 ± 1.69	95.30 ± 1.69
VPgam	84.52 ± 3.05	90.95 ± 2.96	91.43 ± 3.21	92.86 ± 2.96	94.76 ± 3.16	93.81 ± 2.27	92.38 ± 3.01	95.71 ± 1.23	95.48 ± 0.91
VPih	81.37 ± 8.49	89.86 ± 3.64	91.04 ± 3.57	93.63 ± 1.94	95.05 ± 2.36	96.70 ± 1.81	95.05 ± 2.82	95.52 ± 1.61	96.23 ± 0.77
VPii	95.91 ± 1.29	97.63 ± 1.29	97.84 ± 0.50	97.63 ± 0.83	98.92 ± 0.43	99.35 ± 0.83	98.28 ± 0.70	98.71 ± 0.50	98.49 ± 0.83
VPja	89.56 ± 3.91	91.99 ± 3.21	93.45 ± 3.30	93.20 ± 2.38	95.87 ± 2.43	95.87 ± 0.93	95.87 ± 0.93	96.60 ± 2.02	95.15 ± 2.10
VPsaj	94.44 ± 2.62	95.14 ± 2.66	93.98 ± 1.20	95.37 ± 0.76	96.99 ± 1.58	95.60 ± 1.91	96.76 ± 0.93	97.45 ± 1.17	97.69 ± 1.60
VPsal	70.43 ± 7.21	82.69 ± 5.03	80.77 ± 3.42	89.90 ± 2.29	91.11 ± 1.98	93.03 ± 1.98	86.78 ± 5.79	91.83 ± 2.29	91.35 ± 1.36
Average	84.21	90.75	91.97	93.39	95.11	95.10	93.21	95.28	95.38
*p* value	-----	4.54 × 10^−5^	5.71 × 10^−5^	-----	3.90 × 10^−7^	1.71 × 10^−4^	-----	2.43 × 10^−4^	6.69 × 10^−4^

Note: The *p* value is calculated by two-tailed paired *t* tests, where “-----” means not applicable.

**Table 2 brainsci-13-00268-t002:** Classification accuracies achieved by different algorithms on the HGD.

Subject	DeepConvNet			EEGNet			ShallowConvNet		
	Original	TRM-(5,5)	TRM-(3,3)	Original	TRM-(5,5)	TRM-(3,3)	Original	TRM-(5,5)	TRM-(3,3)
S1	57.81 ± 3.70	62.81 ± 3.77	63.91 ± 1.56	54.84 ± 6.28	58.75 ± 3.31	56.72 ± 5.87	67.34 ± 1.72	72.66 ± 1.72	77.81 ± 4.22
S2	70.78 ± 2.19	69.84 ± 2.62	73.91 ± 1.72	71.72 ± 3.44	73.13 ± 2.34	77.34 ± 2.41	79.53 ± 1.56	82.97 ± 1.80	81.25 ± 2.22
S3	83.13 ± 7.09	90.16 ± 1.48	92.03 ± 3.20	92.81 ± 1.57	95.31 ± 0.81	96.72 ± 1.29	95.16 ± 1.07	95.31 ± 0.81	95.00 ± 1.53
S4	84.06 ± 4.64	90.00 ± 2.75	87.34 ± 6.52	93.44 ± 1.88	95.47 ± 1.29	97.50 ± 1.35	96.09 ± 0.60	97.03 ± 1.29	97.81 ± 0.81
S5	62.97 ± 5.16	67.66 ± 5.14	82.50 ± 7.12	70.16 ± 8.53	72.97 ± 4.55	81.88 ± 8.52	81.25 ± 4.89	90.16 ± 1.29	89.06 ± 2.31
S6	57.97 ± 2.19	70.63 ± 5.66	65.47 ± 4.19	74.84 ± 3.83	85.47 ± 1.72	78.28 ± 9.39	86.09 ± 1.39	91.88 ± 0.88	92.03 ± 1.56
S7	59.69 ± 3.48	67.66 ± 4.69	68.28 ± 1.93	65.31 ± 6.05	69.53 ± 4.00	71.41 ± 5.26	75.16 ± 0.94	81.09 ± 5.74	84.38 ± 4.27
S8	72.66 ± 2.00	75.47 ± 2.57	74.69 ± 3.33	75.16 ± 1.39	82.81 ± 5.92	83.75 ± 5.47	81.25 ± 3.78	87.34 ± 3.69	92.19 ± 2.58
S9	47.51 ± 7.98	69.84 ± 4.00	70.00 ± 3.99	76.56 ± 8.50	68.75 ± 6.88	84.84 ± 8.07	78.75 ± 2.10	76.88 ± 2.98	85.63 ± 2.93
S10	78.91 ± 1.39	84.22 ± 0.79	76.09 ± 3.12	84.22 ± 3.80	87.34 ± 2.30	86.88 ± 3.02	85.47 ± 2.99	89.69 ± 0.81	89.22 ± 0.31
S11	62.66 ± 2.99	60.00 ± 5.71	68.28 ± 6.38	73.44 ± 6.30	75.66 ± 7.45	79.38 ± 8.12	78.28 ± 3.97	96.72 ± 1.07	95.31 ± 1.80
S12	80.94 ± 4.69	78.91 ± 0.60	79.69 ± 4.13	82.19 ± 5.27	85.16 ± 2.86	86.88 ± 3.85	88.59 ± 2.86	92.03 ± 1.64	91.09 ± 0.31
S13	59.38 ± 4.59	65.63 ± 3.35	67.81 ± 3.17	75.78 ± 1.18	80.94 ± 4.75	77.81 ± 6.30	79.06 ± 3.77	87.66 ± 1.56	85.47 ± 3.48
S14	60.78 ± 5.96	71.09 ± 5.04	75.78 ± 7.15	80.31 ± 4.10	81.72 ± 8.83	82.19 ± 6.13	73.59 ± 4.96	76.09 ± 3.48	77.34 ± 2.72
Average	67.09	73.14	74.70	76.48	79.50	81.54	81.83	86.97	88.11
*p* value	-----	3.73 × 10^−3^	1.75 × 10^−3^	-----	0.015	1.95 × 10^−5^	-----	1.72 × 10^−3^	2.04 × 10^−4^

Note: The *p* value is calculated by two-tailed paired *t* tests, where “-----” means not applicable.

## Data Availability

The Emergency Braking During Simulated Driving Dataset is available under http://bnci-horizon-2020.eu/database/data-sets (accessed on 1 February 2023), and the High Gamma Dataset is available under https://gin.g-node.org/robintibor/high-gamma-dataset/src/master/data (accessed on 1 February 2023).

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
