# Peer review of "Convolutional Neural Network with a Topographic Representation Module for EEG-Based Brain—Computer Interfaces"

_brainsci, 2023, doi:10.3390/brainsci13020268_

Round 1
Reviewer 1 Report
This study aimed to propose a BCI approach using eeg signals. I have the following suggestions.
What is the novelty of this study although several DL approaches for BCI approach using eeg signals have been proposed earlier? Authors used compare three published method.
Please write down the contribution of the study at the end part of the Introduction section in bulleted form.
EEG is highly sensitive to the powerline, muscular, and cardiac artifacts. In EEG data preprocessing, authors need to mention how you handle AC power, ECG, and EMG artifacts in EEG signals. Same for EOG, EMG and others. Do the authors think that their proposed method is robust to such kinds of artifacts?
Authors should improve the conceptual figures of DL proposed frameworks with more details and model parametrization.
What is the epoch length of EEG signal? 4-125 Hz is very wide range for BCI. It includes EMG, AC power noise and other artifacts. Around 12-30 Hz is most desirable.
Authors should introduce the EEG applications in ML/DL-based disease, and mental workload prediction in broad scope, such as article, Explainable Artificial Intelligence Model for Stroke Prediction Using EEG Signal; in article, healthsos: real-time health monitoring system for stroke prognostics; in article, quantitative evaluation of task-induced neurological outcome after stroke; in article, driving-induced neurological biomarkers in an advanced driver-assistance system; and in article, quantitative evaluation of eeg-biomarkers for prediction of sleep stages.
The authors need to mention the model parameters or hyperparameters of DL models. The performance of the model is dependent on the selection of the architecture and/or parameters.
Authors should report more performance measures of classifiers, such as sensitivity, specificity, precision, and negative predictive value from the confusion matrix.
Both cross-validated training and testing ROC curves.
How did the authors deal with dataset class imbalance challenges in DL analysis?
I recommend to use deepSHAP/Grad-CAM to explain the models.
The discussion section needs to be improved. Authors must make discussion on the advantages and drawbacks of their proposed method with other studies adding a table in the discussion section.
Clinical explanation of these findings needs to be described in support of reference.
From the writing point of view, the manuscript must be checked for typos and the grammatical issues should be improved.
Reviewer 2 Report
Please describe how much missing data was in the datasets and did authors make pre-processing or not.
How did you find the most suitable parameters for your methods? Considering making a subsection to discuss it inside the methodology.
How dataset is divided into training, testing, and validation sets.
Authors should use LOSO metric and confidence intervals to evluate the proposed model.
Reviewer 3 Report
In the paper title “Convolutional Neural Network with a Topographic Representation Module for EEG-Based Brain-Computer Interfaces” introduce an EEG topographic representation module (TRM) and three widely used CNNs 20 (ShallowConvNet, DeepConvNet, and EEGNet) for a classification task.
Although the paper is significant overall, I have some major feedback for the authors.
Major comments:
· Abstract: The authors state that their objective is to create a CNN with the capacity to learn spatial topological characteristics and enhance classification performance while essentially preserving its original structure. The categorization object, however, is not made explicit throughout the entire work. The categorization strategy is also not clear. I would propose incorporating this idea within the paper. It appears from Figure 1 that the authors are preprocessing EEG time information rather than utilizing CNN for categorization. This subject seems confusing.
· Are lines 31 and 32 actually helpful and essential to the paper?
· Introduction: I would recommend improving how the classification aim is addressed.
· Section 2.1 The EBDSDD task remain unclear. There appear to be two categories of tasks (emergency braking and normal driving). According to the section information, how many regular driving tasks are there in addition to the 210 emergency braking (target) trials?
· Line 137-139: The subject of this statement is unclear and confusing. 1500 ms are inter stimuli interval? I advise concentrating on the EBDSDD description. Information about the ground and references may also be offered.
· The mapping block method resembles a 2D to 3D reshape operation together with channel zeroing. This strategy's rationale is vague. What does the W dimension here mean? The sources the writers cite make use of this strategy but do extra research. I propose expanding the mapping block to clarify this strategy.
· Another suggestion I'd like to make is to include information on the size and structure of the entire datasets utilized to train CNNs. In addition to courses, the general (Subject x H x W x TP).
· I'd like to recommend adding a few sentences on the train/test dataset and the validation techniques utilized.
Round 2
Reviewer 1 Report
Authors should report the overall DL results, not according to the subjects.
Authors should interpret the features of DL results using a XAI method.
Data balance is also essential to implement.
Reviewer 2 Report
Authors have addressed my comments
Author Response
Thank you very much for your constructive comments.
Reviewer 3 Report
I have no more comments. I would recommend reordering the references. Line 41 displays [50-54] in the incorrect order; it should read [2-3].
Author Response
Point 1: I would recommend reordering the references. Line 41 displays [50-54] in the incorrect order; it should read [2-3].
Response 1: Thank you for the suggestion. In the revised manuscript, we have revised the citation order of references accordingly.
We appreciate your constructive comments very much and hope that the corrections will be met with your approval.